# Predictive modeling of consumer purchase behavior on social media: Integrating theory of planned behavior and machine learning for actionable insights

**Md. Shawmoon Azad, Shadman Sakib Khan, Rezwan Hossain, Raiyan Rahman, Sifat Momen** [ID]*

Department of Electrical and Computer Engineering, North South University, Dhaka, Bangladesh

* sifat.momen@northsouth.edu

**Data Availability Statement:** The data used can be accessed from https://data.mendeley.com/datasets/nzy5528nst/2 DOI: 10.17632/nzy5528nst.

## Abstract

In recent times, it has been observed that social media exerts a favorable influence on consumer purchasing behavior. Many organizations are adopting the utilization of social media platforms as a means to promote products and services. Hence, it is crucial for enterprises to understand the consumer buying behavior in order to thrive. This article presents a novel approach that combines the theory of planned behavior (TPB) with machine learning techniques to develop accurate predictive models for consumer purchase behavior. This study examines three distinct factors of the theory of planned behavior (attitude, social norm, and perceived behavioral control) that provide insights into the primary determinants influencing online purchasing behavior. A total of eight machine learning algorithms, namely K-nearest neighbor, Decision Tree, Random Forest, Logistic Regression, Naive Bayes, Support Vector Machine, AdaBoost, and Gradient Boosting, were utilized in order to forecast consumer purchasing behavior. Empirical findings indicate that gradient boosting demonstrates superior performance in predicting customer buying behavior, with an accuracy rate of 0.91 and a macro F1 score of 0.91. This holds true when all factors, namely attitude (ATTD), social norm (SN), and perceived behavioral control (PBC), are included in the analysis. Furthermore, we incorporated Explainable AI (XAI), specifically LIME (Local Interpretable Model-Agnostic Explanations), to elucidate how the best machine learning model (i.e. gradient boosting) makes its prediction. The findings indicate that LIME has demonstrated a high level of confidence in accurately predicting the influence of low and high behavior. The outcome presented in this article has several implications. For instance, this article presents a novel way to combine the theory of planned behavior with machine learning techniques in order to predict consumer purchase behavior. This integration allows for a comprehensive analysis of factors influencing online purchasing decisions. Also, the incorporation of Explainable AI enhances the transparency and interpretability of the model. This feature is valuable for organizations seeking insights into factors driving predictions and the reasons behind certain outcomes. Moreover, these observations have the potential to offer valuable insights for businesses in customizing their marketing strategies to align with these influential factors.

2 Code can be found at https://github.com/shawmoonazad/Purchase_Behavior_TPB.

**Funding:** The author(s) received no specific funding for this work.

## Introduction

Social media marketing makes use of platforms to advertise businesses, interact with consumers, and drive business growth. There were 4.48 billion social media users worldwide as of April 2023, which is almost 60% of the world's population [1]. Such massive popularity is due to the comfort and services the social media platforms bring to end users [2, 3]. Moreover, it is anticipated that the e-commerce industry would yield around $3.65 trillion in revenue by 2023, exhibiting a compound annual growth rate of 11.22%. Consequently, the market volume is projected to reach $5.58 trillion by the year 2027 [4]. These statistics clearly demonstrate the rising popularity of social media and consumers' reliance on it, pointing to a clear prospect for social media marketing possibilities for e-commerce businesses globally. Therefore, understanding purchase behavior on social media is crucial for businesses as it helps identify and predict consumer purchasing patterns, preferences, and trends. By leveraging this understanding, businesses can make informed decisions regarding marketing and sales strategies, effectively allocate resources, and maximize return on investment on social media marketing. Furthermore, understanding consumer purchasing behavior provides a competitive advantage in a fiercely competitive marketplace. Analysing these patterns also empowers companies to refine their future marketing strategies and enhance product development initiatives.

Several studies on consumer behavior patterns and decision-making have taken statistical learning theory [5] and machine learning [6–9] approach. Other studies have taken psychological theories, such as the integrated model of behavioral prediction [10], social cognitive theory [11], and theory of planned behavior (TPB) [12–14], into their consideration. This article aims to examine various factors that influence the purchasing behavior of users on social media platforms. Our investigative methodology is primarily data driven. The dataset collected by [15] was utilized for the purpose of our inquiry. The present dataset encompasses a survey that has been constructed based on the theoretical framework of the theory of planned behavior. TPB establishes a connection between attitude, social norm, and perceived behavior control, which ultimately influences the intention that propels the customer's behavior. Machine learning methodology is employed to comprehend the degree to which these factors influence a customer's purchasing decision.

In recent times, social media platforms have emerged as influential instruments for shaping consumer behavior. As the interaction between consumers with these platforms increases at a rapid rate, organizations feel a need for a thorough understanding of the key factors underlying online consumer buying behavior. To gain an in-depth understanding of the current landscape, it is essential to identify the underlying factors that drive consumer buying behavior. This comprehension empowers organizations to customize their marketing plans with high precision, allocate resources effectively, and elevate customer experiences. While conventional models (such as TPB) have predominantly centered around psychological factors or machine learning in isolation, the integration of the TPB with machine learning techniques (and explainable AI) opens up a unique opportunity to construct predictive models that offer a distinct perspective capable of capturing the intricate subtleties of online purchasing decisions.

In this research, we have integrated the theory of planned behavior and machine learning to create models that predict consumer buying behavior with high reliability. Furthermore, the inclusion of XAI delves into the exploration of key influencing factors behind online buying.

The key contributions of this research are summarized as follows:

- The present study employed an integration of the theory of planned behavior with machine learning techniques to gain insights into consumer purchasing behavior on social media platforms.

- The distribution of the dataset utilized in this study was examined in relation to attitude, social norms, perceived behavioral control, and purchase behavior using simple data visualization techniques.

- A total of eight machine learning algorithms were utilized to assess the predictive capabilities of seven combinations of the theory of planned behavior factors (attitude, social norms, perceived behavioral control) in determining purchase behavior.

- Explainable AI (XAI) tool was employed on the best machine learning model to elucidate the black box model which in turn provides a deeper understanding to key factors contributing towards purchasing behavior.

The rest of the paper is organized in the following order. Related background is provided after this introductory section. Following this, we discuss the methodology of the workflow. The result section discusses the results obtained and delves into further investigation. After the results section, we discuss possible managerial and social implications of this research. In the discussion section, we highlight the key achievements and limitation of the work. Finally we conclude the work and discuss future research directions.

## Related works

### Related works on predicting purchase behavior using the theory of planned behavior

Theory of planned behavior has been widely utilized to get insights into consumers' purchasing intentions and behavior. According to Ajzen [16], the theory posits that conduct is influenced by intention, which is shaped by an individual's attitude, subjective norm, and perceived control. The author also delineated the practical applicability of TPB in forecasting individual product preferences as well as general trends in food consumption. Numerous studies have employed the theory of planned behavior to forecast individuals' intentions regarding online grocery shopping [17], online shopping [18], and the selection of shopping centers [19]. These research collectively indicate a prevailing agreement that attitude, subjective norms, and perceived behavioral control exert significant influence on both intentions and purchasing behavior. Ketabi et al. [20] conducted an investigation to examine how factors impact the purchasing decisions of online consumers. They employed a conceptual framework to investigate the influence of many elements, such as subjective norms, behavioral control, attitude, the role of friends, and perceived credibility, on purchasing intentions. In his study, George [14] investigated the correlation between individuals' views of behavioral control, reliability, and online privacy. His research demonstrated that trustworthiness has a favorable impact on individuals' attitudes towards online purchase—thus leading to positive effects on customer behavior. Lim and Dubinsky [21] conducted a thorough examination of belief constructions that included interdependency elements inside salient beliefs. They were able to provide precise explanation for customers' intents to engage in online purchasing. Pavlou and Fygenson [22] conducted a study that investigated the adoption of electronic commerce by customers. The researchers employed the theoretical framework of TPB to explore the association between two interrelated online activities and the accompanying intentions. In their study, Pelling and White [23] built upon the existing theoretical framework proposed by TPB and introduced additional variables, namely self-identity and belongingness, in order to enhance the predictive capability of consumer behavior. Shin et al. [24] found that the variables of the Theory of Planned Behavior (TPB), namely attitude, subjective norm, and perceived behavioral control, were able to effectively predict the intention to purchase state-branded food products. Furthermore, the study

also determined that there was no significant correlation between health consciousness and intention. Widyarini et al. [25] discovered that along with TPB variables, namely attitude and perceived behavioral control, the addition of self-determination and motivation exhibited predictive capabilities in determining the intentions of individuals to purchase online fashion products. However, subjective norms had little or no influence on the prediction of purchasing intentions. Bangun et al. [26] also reported comparable results. Economie [27] conducted a comparative analysis of the Theory of Planned Behavior (TPB) and the Relationship Quality (RQ) models, revealing that the TPB variables exhibited more efficacy in predicting intention. Hansen et al. [17] also discovered TPB to outperform the Theory of Reasoned Action (TRA) in predicting online grocery shopping intention. In a recent study conducted by Han et al. [13], a comprehensive meta-analysis was performed to examine the relationship between attitude, subjective norm, perceived behavioral control, and intention in the context of socially responsible consumer behavior. The findings of this study revealed that attitude, subjective norm, and perceived behavioral control were significant predictors of intention.

## Related works on predicting purchase behavior using machine learning

In recent years, researchers have utilized machine learning approaches to predict consumer purchasing behavior. Ebrahimi et al. [7] used structural equation modelling and unsupervised machine learning approaches to demonstrate how social media marketing can affect consumer purchase behavior. The study was based on five factors (e.g., entertainment, customization, interaction, word of mouth and trend) of social networks that influence consumer purchase behavior. To build their dataset, they used convenience sampling approach and common bias method and did a survey on 466 respondents. Chaubey and colleagues [28] used wholesale customers' data from the UCI-Irvin machine learning data repository, which contained 440 samples, to compare various supervised machine learning classification algorithms and hybrid algorithms. They found that the best classifier was the hybrid classifier using the ensemble stacking technique which attained an accuracy of 92.42%. Quynh and Dung [29] made predictions regarding the likelihood of a client accepting a coupon for a particular venue. These predictions were based on a consideration of demographic and contextual factors. Highest accuracy was achieved in Random Forest and Bagging classifier, attaining an accuracy of 76%. Choudhury and Nur [6] predicted potential customer based on purchase behavior. They used a dataset which was collected from a grocery superstore named "Taradin Super Shop". The dataset contained 9259 customers sales data. After the experiment was done, it was observed that 43% were potential customers and 57% were non-potential customers. Five different classifiers were applied, and the best performance was attained in logistic regression 98.49%. Do and Trang [30] used the machine learning approach to forecast Vietnamese consumers' purchase behavior. They used a very small dataset of Vietnamese consumers of 2018 which contained 240 samples. The dataset was preprocessed and trained using four different machine learning algorithms, and it was observed that decision tree attained the highest accuracy 91.67%. Carreón et al. [31] attempted to measure the mere exposure effect and perceptual fluency effect of television advertisements on predicting purchasing behavior using a dataset which contained 3000 customers across 36 different products during the span of 3 months. They used three classifiers (namely SVM, XGBoost and logistic regression) to measure the influence. The researchers discovered that predictive models relying on the duration of exposure to television advertisements exhibit notably diminished accuracy in forecasting actual purchase behavior when compared to models that incorporate demographic information. Juárez et al. [32] employed an adaptive machine learning approach to conduct Neuromarketing Analysis and forecast consumer purchase behavior. Their dataset contained 375 records. The

findings imply that the package's graphic aspects, which the approach thoroughly evaluates and categorizes based on social context and the type of consumer that is looking at the packaging, are its most crucial components. Two approaches namely, the statistical approach and machine learning approaches, were taken by Safara [33] to predict consumer behavior in an e-commerce environment in the COVID-19 era. The data was obtained from the DigiKala website. A correlation analysis was conducted on the dataset. Subsequently, the dataset underwent preprocessing and was trained utilizing a machine learning algorithm, with the ultimate selection of the most optimal model. The decision tree ensembles exhibited the highest level of performance, achieving an accuracy rate of 95.3%. Rubi et al. [34] conducted a comprehensive investigation to assess the purchasing behavior of consumers regarding travel insurance. The researchers utilized a dataset comprising 3980 instances of individuals who travel and proceeded to train this dataset using 10 distinct classifiers. This study aimed to investigate these individuals' purchasing patterns and develop a predictive model that can determine the likelihood of a consumer purchasing insurance, among the various classifiers namely Random Forest, Decision Tree Classifier, and Stochastic Gradient Descent.

## Research gap in literature

Despite the existence of numerous studies that have independently applied the theory of planned behavior and machine learning techniques to forecast consumer behavior, it is important to acknowledge the presence of significant research gaps within the current body of literature. The following are some important research gaps we have identified:

- **Integration gap**: The current body of research is deficient in examining effective integration of the psychological elements of TPB with machine learning techniques for the purpose of predicting online customer behavior.

- **Model interpretability gap**: The model interpretability gap refers to the need to not only forecast consumer purchasing behavior but also to grasp the underlying causes behind these actions in order to acquire a more comprehensive understanding. Explainable artificial intelligence has distinct prospects for delving into this area of inquiry. While prior studies have examined the statistical methodology (see Table 1) for assessing the diverse effects of various

**Table 1. A comparative analysis of related works on predicting purchase behavior using machine learning.**

| REF | Subject | Sample size | Data collection means | Advantage | Limitation |
|---|---|---|---|---|---|
| [7] | People living in Hungary using Facebook Marketplace | 466 | Survey dataset based on convenience sampling approach and common bias method. | Investigates social media influence on consumer purchase behavior | Few machine learning algorithms |
| [28] | HoReCa customers and retail customers | 440 | Wholesale customers' data from the UCI-Irvine machine learning data repository. | Multiple machine learning approaches were taken to verify their study. | Lack of Explainable AI |
| [29] | Car drivers and passengers | 12,684 | Amazon Mechanical Turk poll published at UCI website. | Large dataset was used in this study | Low accuracy |
| [6] | Customers of "Taradin Super Shop" | 9,259 | Grocery superstore named "Taradin Super Shop" provided the sales data | Large dataset was used | The area of the research only limits to the grocery superstore |
| [31] | Customers of TV adverts | 3,000 | Nomura Research Institute Ltd. | Investigates the influence of TV commercial adverts on purchase behavior | Limited number of machine learning models used |
| [32] | Children and Parents | 375 | Gazepoint Analysis | Neuro market analysis was used | Very small dataset was used |
| [34] | Travelers | 3,980 | Travel insurance | Investigates consumer behavior based on travel data | Low accuracy |

components on overall prediction, the utilization of XAI has not been widely documented in scholarly literature.

This article has looked into these two areas which have predominantly been overlooked in the current body of knowledge.

## Methodology

Fig 1 illustrates the methodology adopted in this study. The dataset utilized in our study is taken from [15]. The TPB-related features were retrieved from the dataset, followed by the execution of appropriate data preprocessing techniques. Subsequently, data visualization was conducted in order to examine the relationships between data and derive insights based on the data. Following this, the dataset was partitioned into two subsets: a training set including 80% of the data, and a test set comprising the remaining 20%. Subsequently, machine learning techniques were employed on the training set in order to construct models, and their efficacy was

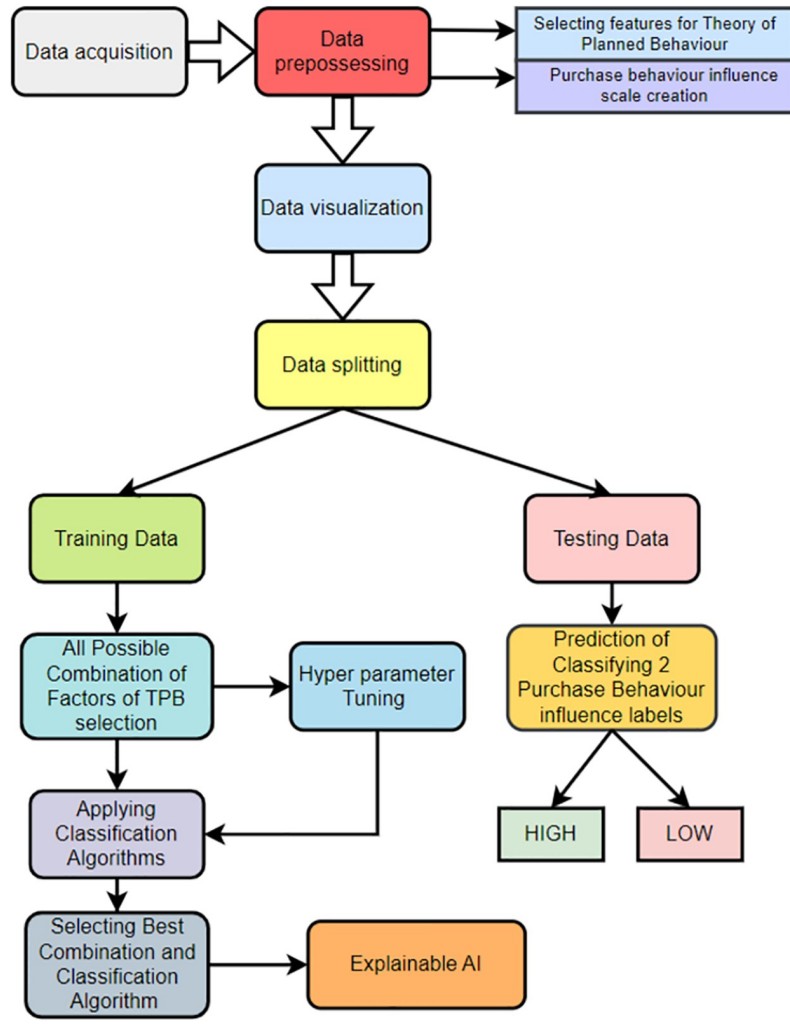

**Fig 1. Methodology of the study.**

evaluated by means of the test set. Finally, we employed Explainable AI techniques on the best model to interpret the predicted outcomes.

## Data acquisition

The survey dataset was obtained from the study conducted by [15], wherein a total of 219 participants provided complete responses. There are multiple rationales that support the appropriateness of utilizing survey-based data collection as a strategy. Firstly, in their study, Zhou et al. [15] developed a meticulously structured questionnaire with a specific set of inquiries, with the intention of obtaining responses solely to these predetermined questions. Survey based strategy suits the best for this circumstances. Secondly, surveys offer the advantage of efficiently gathering data from a large sample size. Furthermore, surveys allow valuable insights into participants' perspectives. Anonymity, facilitated by surveys, serves to mitigate common method bias.

The distribution of questionnaires was conducted through the utilization of Google forms, targeting social media users residing in Malaysia. The researchers employed a snowball sampling methodology in order to enlist individuals for the study. Snowball sampling is a non-probabilistic technique employed for the purpose of gathering data from individuals who possess some particular traits that researchers find relevant to their study. The initially recruited participants, subsequently, proceed to recruit further subjects for the purposes of the study. Conducting a study on the experience and influence of online purchase would be irrelevant if non-internet users were included. Therefore, the authors focused their interest on individuals who use internet actively. Furthermore, the authors restricted the scope of their investigation to Malaysia, thereby focusing on another specific demographic characteristic. The authors used the popular social media platform,WeChat, to distribute the link to the Google Form among Malaysian online users.

The questionnaire comprised a total of 26 questions, which were organized into two distinct sections—Section A and Section B. Section A and B consisted of 10 and 16 questions respectively. The questions from section A were pertaining to the demographic characteristics of the participants (see Table 2 for the demographic profile of respondents) as well as some basic questions related to the experience of the social media usage of participants. On the other hand, questions from section B were primarily aimed at understanding the perception of factors that influence participants' intention to purchase as well as to make online purchase decisions. Responses to questions in section B were required to be chosen from a 5-point Likert scale, which encompassed a range from 1, indicating "Strongly Disagree" to 5, indicating "Strongly Agree".

Figs 2 and 3 shows the occupation of the participants across gender and ethnicity, respectively.

## Common method bias

Common method bias (CMB) refers to bias that arises when both the independent and dependent variables are gathered through the same survey instrument or response method. This results in distortion in response that comes from the process rather than the respondent's inclination. Various strategies to mitigate CMB have been implemented throughout the data collection process in [15]. Firstly, it is important to note that the survey in its entirety ensured anonymity to participants, which facilitates their ability to respond without feeling compelled to provide socially desired responses. In addition, the questions posed were succinct and precise—thus eliminating any potential ambiguity or uncertainty among the participants. Furthermore, the brevity of the survey facilitates quick completion for respondents, minimizing

**Table 2. Demographic profile of respondents.**

| Demographic Variables | Category | Frequency | Percentage (%) |
|---|---|---|---|
| Gender | Male | 99 | 45.20 |
| | Female | 120 | 54.80 |
| Age | 17—22 | 45 | 20.55 |
| | 23—28 | 69 | 31.51 |
| | 29—34 | 54 | 24.65 |
| | 35—40 | 39 | 17.80 |
| | Others | 12 | 5.49 |
| Ethnicity | Malay | 21 | 9.59 |
| | Chinese | 180 | 82.19 |
| | Indian | 17 | 7.76 |
| | Others | 1 | 0.46 |
| Annual Income | Less than RM 30,000 | 40 | 18.27 |
| | RM 30,001—RM 50,000 | 51 | 23.29 |
| | RM 50,001—RM 70,000 | 50 | 22.83 |
| | RM 70,001—RM 90,000 | 27 | 12.32 |
| | RM 90,001 and above | 51 | 23.29 |
| Occupation | Student | 57 | 26.03 |
| | Businessman | 49 | 22.37 |
| | Homemaker | 16 | 7.31 |
| | Employee | 94 | 42.92 |
| | Retired | 3 | 1.37 |

the time required for their participation. As a result the potential for bias arising from boredom, which may be caused by a longer period to finish the survey, is reduced. These attributes allowed the data procurement process to minimize any effect that may have altered the data due to biases such as common method bias.

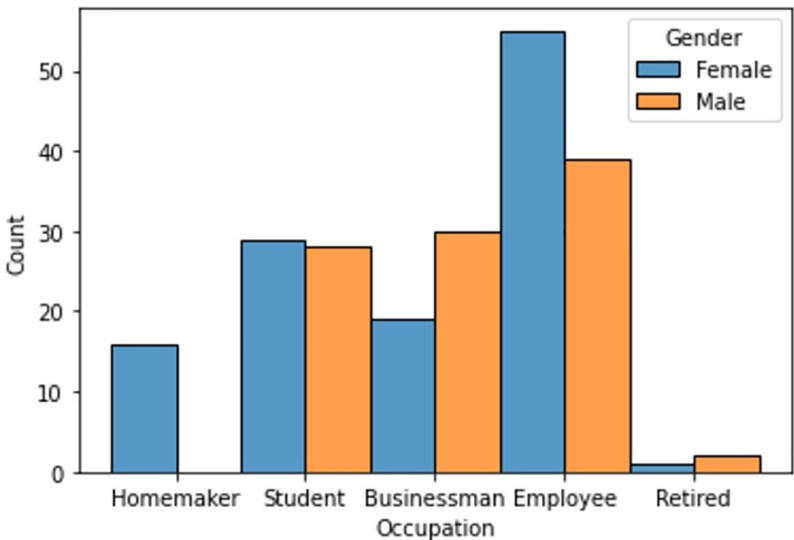

**Fig 2. Gender-wise distribution amongst participants.**

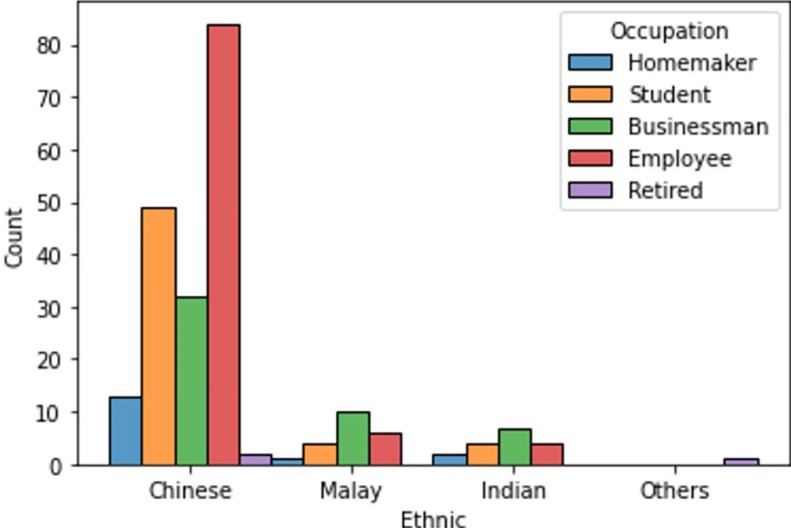

**Fig 3. Distribution of occupation across different ethnic group.**

## Non response bias

Non-response bias refers to the phenomenon when individuals who are selected to participate in a survey exhibit an unwillingness or inability to provide a response to either a specific survey question or the survey as a whole. The factors contributing to non-response are diverse and can differ among individuals. A commonly employed method for assessing the presence of non-response bias in survey data involves comparing the responses of individuals who responded early to those who responded later. Late respondents might be conceptualized as individuals who had a lower level of willingness to engage and hence provided their responses at a later point in time. In order to ascertain the presence of non-response bias within the survey data, a comparative analysis was conducted between the first 50 respondents (referred to as the "early" group) and the last 50 respondents (referred to as the "late" group). This analysis was performed using the paired samples t-test method in the Statistical Package for the Social Sciences (SPSS) software, version 29. The results of our study suggest that of the 16 variables examined (see Table 4), only 3 exhibited statistically significant disparities (with a two-tailed p-value below 5%) in the average responses between the two groups of participants. The results are shown in Table 3. While these items are statistically different, the differences are quite small and generally would not affect the overall interpretation of the results.

## Data prepossessing

**Selecting features for theory of planned behavior.** TPB posits that purchase behavior is influenced by the variables of attitude, social norms, and perceived behavioral control.

**Table 3. Statistically significant differences between early and late respondents.**

| Variable | Mean (First 50) | Mean (Last 50) | Statistical Significance |
|---|---|---|---|
| SN1 | 2.66 | 3.26 | 0.022 |
| SN2 | 2.58 | 3.20 | 0.023 |
| PBC4 | 2.90 | 3.42 | 0.039 |

**Table 4. Selected TPB factors and definitions with respective feature set.**

| TPB Factors | Definitions | Feature Set |
|---|---|---|
| Attitude (ATTD) | Attitude pertains to the extent to which an individual possesses a positive or negative assessment of the behavior under consideration. | ATTD1: Social media advertisements can assist me to learn about the existence of the product. ATTD2: Compared to other advertising platforms, social media advertisements can more easily get my attention. ATTD3: I will look for more production-related information if prominent keywords like promotion and discount are used on social media. ATTD4: I have previously engaged in the acquisition of a product that came to my attention via social media. |
| Social Norms (SN) | Social norms refer to the societal expectations and moral beliefs that exert social pressures on individuals, influencing their behavior and actions. | SN1: My family influence my purchasing decision towards social media marketing. SN2: People around me believe that I should purchase on social media. SN3: It makes me happy if many individuals use social media to make purchases. SN4: My friends encourage me to purchase through social media. |
| Perceived Behavioral Control (PBC) | Perceived behavioral control pertains to an individual's subjective assessment of the level of ease or difficulty associated with executing a particular behavior. | PBC1: Frequent advertisement of a product on social media led me to purchase it. PBC2: I will use social media as a purchasing reference channel in the future. PBC3: I will recommend my friends to use social media as a reference when deciding what to buy in the future. PBC4: I will recommend my family to use social media as a reference when deciding what to buy in the future. |
| Purchase Behavior (PB) | Purchase behavior refers to a person's willingness to buy a product. | PB1: I am willing to purchase a social media-marketed product. PB2: There is a high likelihood that I would purchase a product due to social media's influence. PB3: I am readily influenced by social media advertisements and subsequently engage in purchasing behavior. PB4: As a result of social media's influence, I had the experience of purchasing a product. |

Consequently, only the variables linked to these factors were retained, while the remaining variables were excluded. Each of the selected features are posed as statements in 5 point Likert scale to participants and participants have to select the degree to which they agree/disagree to the statements. It is important to note that an excessive quantity of attributes can result in suboptimal outcomes, particularly when considering the constrained size of the dataset. Consequently, the selection of features was determined by taking into account the factors outlined in the theory of planned behavior.

The objective of data pre-processing is to eliminate inconsistencies and convert data into a format that facilitates effective model training, thereby enhancing overall performance. To accomplish this, categorical values present in the feature sets PB1, PB2, PB3, PB4, ATTD1, ATTD2, ATTD3, SN1, SN2, SN3, SN4, PBC1, PBC2, PBC3 (corresponding features from Table 4) have been transformed into numerical values through label encoding. The specific mappings for this encoding process can be found in Table 5.

**Purchase behavior influence scale creation.** After replacing the values with their numerical equivalents, the PB1, PB2, PB3, and PB4 features were merged into a single column named PB (purchase behavior). The maximum value for PB obtained was 20, and the lowest was 4. Finally, we classify the value ranging from 4-12 as LOW class and 13-20 as HIGH class.

**Table 5. Categorical values encoded with corresponding numerical values.**

| Categorical Value | Numerical Value |
|---|---|
| Strongly Disagree | 1 |
| Disagree | 2 |
| Neutral | 3 |
| Agree | 4 |
| Strongly Agree | 5 |

## Data visualization

Fig 4 illustrates the manner in which individuals who are being surveyed responded to inquiries regarding their disposition towards purchasing a product through social media platforms. A majority of respondents, specifically 53.4%, expressed agreement with the notion that social media platforms serve as a valuable tool for acquiring knowledge about the availability of various products. 47% of their responders feel that advertisements on social media has greater impact in getting attention compared to other advertising platforms. Approximately 49% of respondents expressed their inclination to seek additional information regarding a product when keywords related to promotions and discounts are employed in social media. Half of the responders (50.3%) indicated that they had previously participated in the procurement of a product that had been brought to their attention via social media.

Fig 5 illustrates the responses of participants in relation to inquiries regarding social norms. Approximately 41% of the survey participants indicated that their purchasing decisions on social media platforms are influenced by their family members. A comparable proportion of respondents experience a sense of happiness when a significant number of individuals engage in purchasing products through social media platforms. The responses of participants in relation to inquiries regarding perceived behavioral control are depicted in Fig 6. Approximately 40% of the participants acknowledge that the frequent promotion of a product on social media influenced their decision to make a purchase. A total of 42.4% of the responders indicated their intention to recommend the use social media as a reference for their family members when making purchasing decisions.

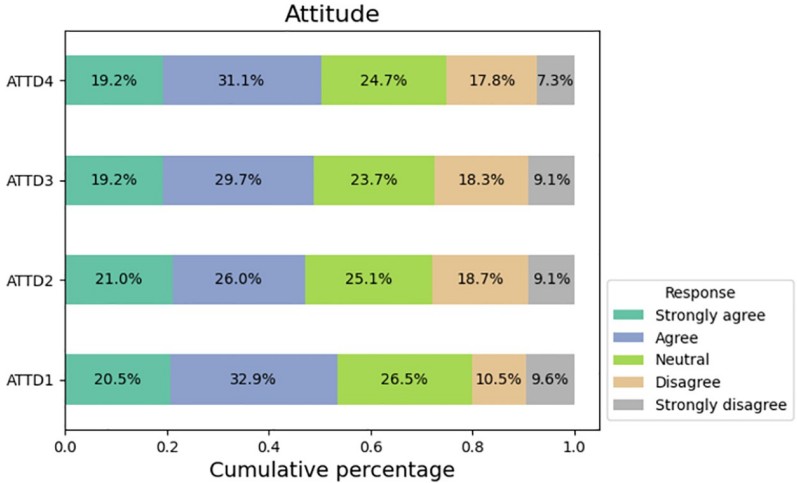

**Fig 4. Responses from attitude factors.**

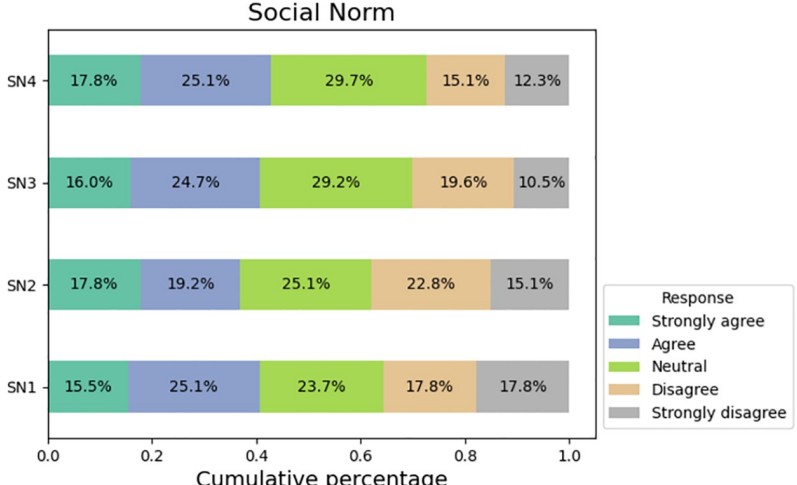

**Fig 5. Responses from social norm factors.**

The participants' responses to inquiries about their purchase behavior are illustrated in Fig 7. According to the survey findings, approximately 41% of respondents express a willingness to engage in the purchase of a product that has been marketed through social media channels. Approximately 51% of the respondents indicated a strong propensity to make a purchase as a result of the influence exerted by social media on the product in question. The survey additionally revealed that approximately 50% of respondents reported having engaged in product purchases as a result of social media influence.

The gender distribution among respondents, as shown in Fig 8(a), reveals that 54.8% of the respondents identified as female, while 45.2% identified as male. From the boxplotIn Fig 8(b), it can be observed that the lower quartile for males' and females' purchase behavior (PB) scores was 8 and 10 respectively. The median score for male participants was approximately 13, whereas for female participants 14.5. This finding suggests that women exhibit a higher propensity to conform to high purchase behavior influence compared to men.

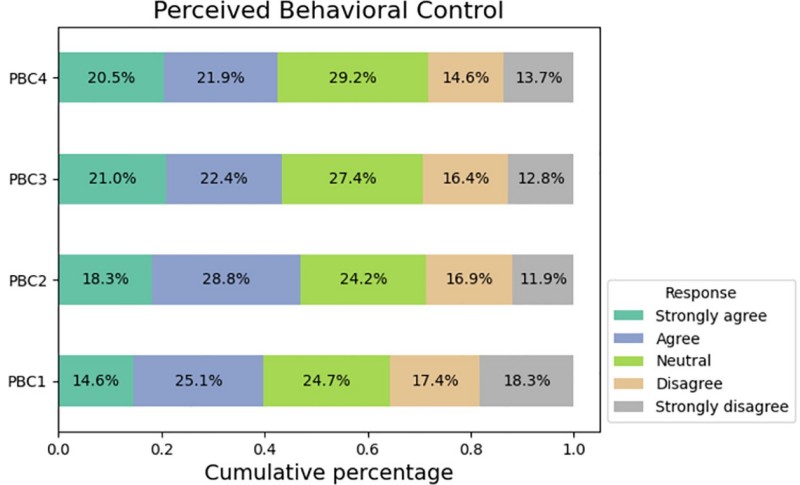

**Fig 6. Responses from perceived behavioral control(PBC) factors.**

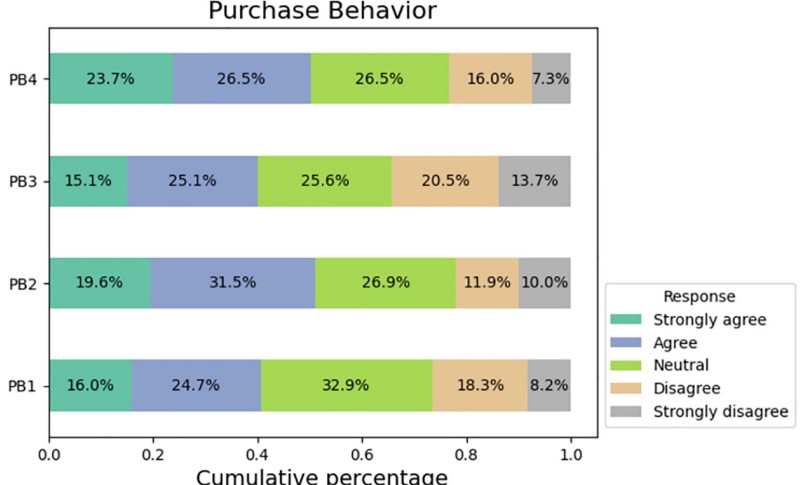

**Fig 7. Responses from purchase behavior factors.**

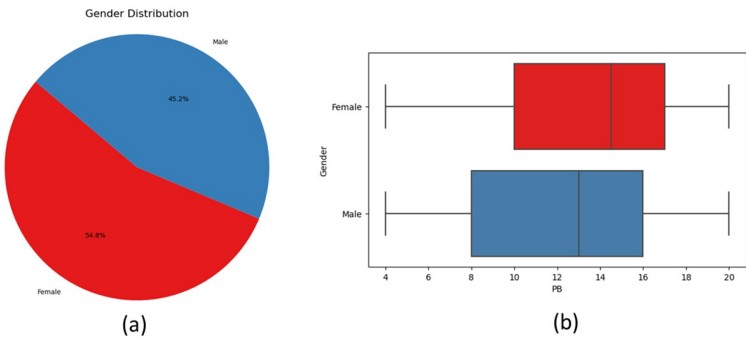

**Fig 8. Percentage distribution(a) of respondent on purchase behavior(b) by gender.**

Fig 9(a) demonstrates the age distribution of the respondents. A total of 31.5% of participants fell within the age range of 23 to 28 years, while 24.7% between 29 and 34 years. Fig 9(b) tells us about the median age against purchase behavior(PB) score. The median PB score for all ages within 23—40 years old were within 14-15, while it was 12 for 17—22 years old people.

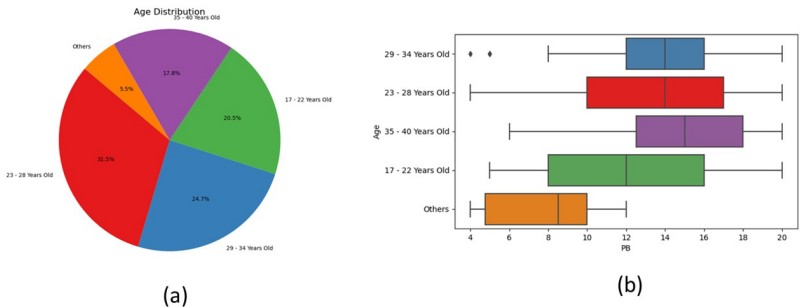

**Fig 9. Percentage distribution(a) of respondent on purchase behavior(b) by age.**

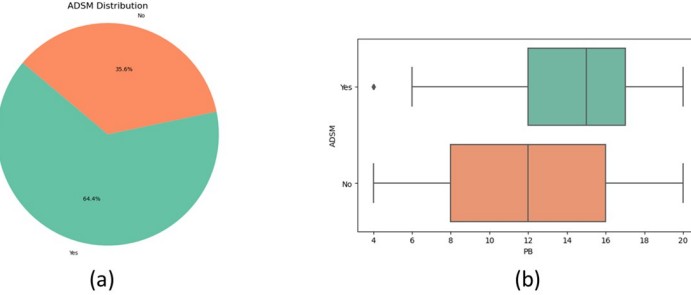

**Fig 10. Percentage distribution(a) of respondent on purchase behavior(b) by attention to the advertisement on social media(ADSM).**

Although both were in the range of high PB score, people around 17—22 years of age had comparatively lower PB score, indicating the fact that these respondents were less influenced by social media in general. The fourth category, which includes minors and people over the age of 40 had an even lower PB score, indicating their much stronger resistance to social media influence.

Fig 10(a) demonstrates how many respondents pay attention to adverts on social media, where almost 64.4% of people said yes and the rest said no. The box-plot Fig 10(b) shows all the people who pay attention to the adverts, are more prone (Median PB score around 15) to influence through social media in terms of purchasing a product. On the other hand, people who do not pay attention to adverts are less likely (Median PB score 12) to get influenced by social media compared to people who pay attention.

## Data splitting

Hold-out validation approach was used to divide the dataset into two distinct sets: the train set and the test set.

- The train set consisted of 80% of the data.

- The test set was the remaining 20% of the data.

The train set was used in the model building phase whereas the model was evaluated on the unseen test set.

## Machine learning algorithms

**K-nearest neighbour classifier.** K-nearest neighbour (KNN) [35] predicts the label of a new, unknown sample based on the labels of the K training samples that are most similar to it. Nearest training samples are obtained using distance metrics such as Euclidean distance or Minkowski distance. For classification tasks, KNN is straightforward and efficient, but it can be computationally expensive and sensitive to the magnitude of the features. The equation of Minkowski distance is given by,

$$d(x, y) = \left( \sum_{i=1}^{n} |x_i - y_i|^p \right)^{\frac{1}{p}} \tag{1}$$

**Decision tree.** A decision tree classifier [36] constructs a model of decisions that resembles a tree, depending on particular features provided by the data. Selecting the path through

the tree that leads to the classification. It can determine the class or category of a given data point. The decision tree employs a top-down methodology. In a decision tree, each decision node is selected based on popular methods, such as Information gain. The equation of information gain is given by,

$$\text{Gain}(D, A) = \text{Entropy}(D) - \sum_{v \in \text{Values}(A)} \frac{|D_v|}{|D|} \times \text{Entropy}(D_v) \tag{2}$$

Here, Eq (2) shows that Gain(D, A) is the information gain of attribute A in Dataset D.

**Logistic regression.** Logistic Regression [37] utilizes the Linear Regression concept to solve classification problems using a plane. To solve the classification problem, it uses the sigmoid function on linear regression. It is used to classify binary classes, such as 0 or 1, true or false, or positive or negative.

**Naive bayes classifier.** Gaussian Naïve Bayes [38] is a classification technique that uses the Bayesian theorem and Gaussian normal distribution to predict the class of a new feature set. The Bayesian formula as follows:

$$P(C|X) = \frac{P(X|C) \cdot P(C)}{P(X)} \tag{3}$$

Here, Eq (3) shows Bayesian formula for two events C and X.

**Support vector machine classifier.** Support Vector Machine is a machine learning algorithm for classifying linear and non-linear data. It finds a hyperplane in high-dimensional space that maximally separates different classes and uses kernels to transform the data into a higher-dimensional space for better separation [39]. SVM aims to classify data points accurately while maximizing the margin between different classes.

**Random forest classifier.** Random forest is an ensemble classifier which employs a collection or combination of several base models such as decision trees [40]. There are two variations of the ensemble technique: (i) bagging (also known as bootstrap aggregation). (ii) boosting. In bootstrapping, the test data is feed to the base models using row sampling with replacement to predict the class. The outcome is based on the base models' majority vote on the test data.

**AdaBoost classifier.** AdaBoost is an ensemble technique that combines weak learners to improve their performance. It is a meta-estimator that initially fits a classifier on the original dataset, then fits other copies of the classifier on the same dataset, with the weights of instances that were incorrectly classified being changed so that subsequent classifiers focus more on difficult instances [41].

**Gradient boosting classifier.** Gradient boosting is an ensemble technique also used in regression and classification tasks [42]. It gives a prediction model as an ensemble of weak prediction models, such as decision trees, to create a strong predictive model.

## Hyperparameter optimization with GgridSearchcv

Hyperparameters are points of choice or configuration that allow a machine-learning model to be customized for a specific task or dataset. GridSearchCV involves defining a search space as a grid consisting of various hyperparameter values. It proceeds by systematically evaluating each position within the grid to determine the optimal combination of hyperparameters. This method allows for a thorough exploration of the parameter space and aids in identifying the best set of hyperparameters for a given machine learning model or algorithm.

## Result analysis

This section presents the performance evaluation of the classifiers utilized in training our dataset. Various metrics, including the confusion matrix, ROC curve, precision-recall curve, and the learning curve of Decision Tree, KNN, Random Forest, Naive Bayes, Logistic Regression, Support Vector Machine (SVM), AdaBoost, and Gradient Boosting classifiers, were made. Accuracy, precision-recall and F1 score were used to measure the performance of these machine learning algorithms on all possible combinations of the Theory of planned behavior (TPB) factors.

Accuracy:

$$\text{Accuracy} = \frac{\text{All Correct Prediction}}{\text{Total Prediction}} \tag{4}$$

Recall:

$$\text{Recall} = \frac{\text{All Correct Positive Prediction}}{\text{Total Actual Positive}} \tag{5}$$

Precision:

$$\text{Precision} = \frac{\text{All Correct Positive Prediction}}{\text{All Predicted Positive Values}} \tag{6}$$

F1-Score:

$$\text{F1} - \text{score} = \frac{2 * recall * precision}{recall + precision} \tag{7}$$

## Hyperparameter optimization

Hyperparameter optimization technique has been used to find the best parameter for a specific model on the dataset. The GridSearchCV hyperparameter optimization technique has been applied to find the optimal parameter values from a given set of parameters in a grid and then used them on the classifier models shown in Table 6.

**Table 6. Parameter space and best parameters of models.**

| Model | Parameter Space | Best Parameters |
|---|---|---|
| k-nearest neighbor | n-neighbors: 1-11 | n-neighbors: 5 |
| Decision Tree | Max_depth: 1-11; Criterion: gini, entropy | Criterion: entropy, Max_depth: 53 |
| Random Forest | Criterion: gini, entropy; Max_depth: 1-11, n_estimators: 10, 50, 100, 200 | Criterion: gini, Max_depth: 6, n_estimators: 100 |
| Logistic Regression | C: logspace(-3,3,7), Solver: newton-cg, lbfgs, liblinear, sag, saga | C: 0.1, Solver: newton-cg |
| Naive Bayes | - | - |
| Support Vector Machine | C: logspace(-3, 3, 7); Kernel: linear, poly, rbf, sigmoid; Gamma: scale, auto | C: 0.1; Gamma: scale; Kernel: linear |
| AdaBoost | n_estimators: 10, 50, 100, 200; learning_rate: 0.001, 0.01, 0.1, 1 | n_estimators: 50, learning_rate: 0.1 |
| Gradient Boosting | n_estimators: 10, 50, 100, 200; learning_rate: 0.001, 0.01, 0.1, 1; max_depth: 1-11 | n_estimators: 200; max_depth: 2; learning_rate: 0.01 |

## Combined performance evaluation of all possible combination of TPB factors

The study demonstrates the performance of various machine learning classification models across different Theory of Planned behavior (TPB) factors. The factors under consideration include Attitude (ATTD), Social Norm (SN), and Perceived behavioral Control (PBC), individually and in their various combinations. The accuracy of these models, derived using Eq (4), is depicted in Table 7, while Table 8 delineates the macro F1 score obtained through Eq (7). Through the integration of eight machine learning algorithms with seven Theory of Planned behavior (TPB) factor combinations, a total of 56 results were obtained. Among these outcomes, the most optimal models were carefully chosen for each specific combination. Subsequently, the superior model and its corresponding combination were identified based on their performance in accuracy and macro F1 Score. The study was focused on predicting two classes characterized by a slightly imbalanced distribution. There were 121 instances associated with the HIGH class, compared to the 98 instances linked to the LOW class. Despite this minor imbalance, the performance outcomes in terms of accuracy and macro F1 score were remarkably equivalent. This implies that the machine learning models were effective in accurately classifying both the HIGH and LOW classes at relatively similar levels. Assessing the performance of individual factors revealed the following: The Gradient Boosting model emerged as the highest performer after examining the combination of ATTD, SN, and PBC as predictors. This model yielded an accuracy and a macro F1 score of 0.91. For ATTD, the AdaBoost and Gradient Boosting classifiers were the top performers, achieving the highest accuracy and

**Table 7. Accuracy table of purchase behavior (PB) on all possible combinations.**

| Model | ATTD & SN & PBC | ATTD | SN | PBC | PBC & ATTD | SN & ATTD | SN & PBC |
|---|---|---|---|---|---|---|---|
| KNN | 0.84 | 0.84 | 0.72 | 0.77 | 0.86 | 0.89 | 0.77 |
| Decision Tree | 0.82 | 0.82 | 0.70 | 0.77 | 0.82 | 0.82 | 0.77 |
| Logistic Regression | 0.84 | 0.84 | 0.84 | 0.80 | 0.86 | 0.86 | 0.80 |
| Naive Bayes | 0.86 | 0.84 | 0.84 | 0.84 | 0.86 | 0.86 | 0.84 |
| SVM | 0.86 | 0.86 | 0.77 | 0.82 | 0.84 | 0.86 | 0.79 |
| Random Forest | 0.89 | 0.84 | 0.77 | 0.77 | 0.85 | 0.84 | 0.77 |
| Ada Boost | 0.86 | 0.86 | 0.72 | 0.70 | 0.84 | 0.86 | 0.75 |
| Gradient Boosting | **0.91** | 0.86 | 0.77 | 0.77 | 0.84 | 0.86 | 0.73 |

ATTD represents Attitude, SN represents Social Norm and PBC represents Perceived Behavioral Control.

**Table 8. Macro F1 scores on all possible combinations.**

| Model | ATTD & SN & PBC | ATTD | SN | PBC | PBC & ATTD | SN & ATTD | SN & PBC |
|---|---|---|---|---|---|---|---|
| KNN | 0.84 | 0.84 | 0.72 | 0.77 | 0.86 | 0.89 | 0.77 |
| Decision Tree | 0.82 | 0.82 | 0.70 | 0.77 | 0.82 | 0.82 | 0.77 |
| Logistic Regression | 0.84 | 0.84 | 0.84 | 0.80 | 0.86 | 0.86 | 0.80 |
| Naive Bayes | 0.86 | 0.84 | 0.84 | 0.84 | 0.86 | 0.86 | 0.84 |
| SVM | 0.86 | 0.86 | 0.77 | 0.82 | 0.84 | 0.86 | 0.79 |
| Random Forest | 0.89 | 0.84 | 0.77 | 0.77 | 0.85 | 0.84 | 0.77 |
| Ada Boost | 0.86 | 0.86 | 0.72 | 0.70 | 0.84 | 0.86 | 0.75 |
| Gradient Boosting | **0.91** | 0.86 | 0.77 | 0.77 | 0.84 | 0.86 | 0.73 |

ATTD represents Attitude, SN represents Social Norm and PBC represents Perceived Behavioral Control.

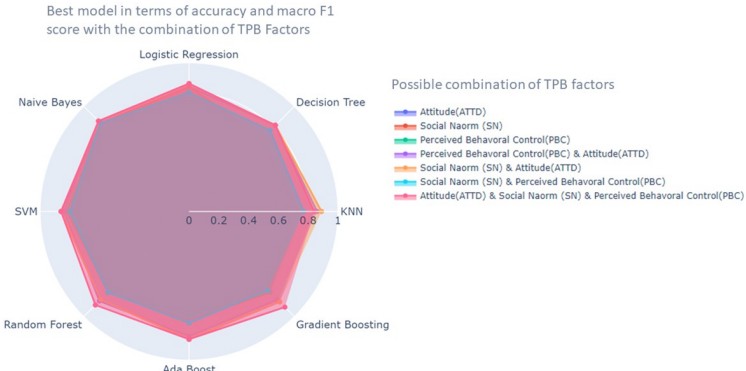

**Fig 11. Radar chart for the best model in terms of accuracy and macro F1 score.**

macro F1 score of 0.86. In regard to SN, the Logistic Regression and Naive Bayes classifiers attained the highest accuracy and macro F1 score, both recorded as 0.84. The Naive Bayes classifier also excelled in PBC, generating the maximum accuracy and macro F1 score of 0.84. When PBC and ATTD were combined as predictors, the K-Nearest Neighbors (KNN) classifier, along with the Logistic Regression and Naive Bayes classifiers, achieved the highest accuracy and macro F1 score of 0.86. In the case of SN and ATTD, the KNN classifier was the most accurate, yielding the highest accuracy and macro F1 score of 0.89. Lastly, the Naive Bayes classifier, when employed with SN and PBC, delivered the highest accuracy and macro F1 score of 0.84. Therefore, upon analyzing the different machine learning models with different combinations of TPB factors, it is evident that Gradient Boosting yielded the highest accuracy and macro F1 score of 0.91, thus outperforming all other models.

Fig 11 depicts the outstanding performance of the Gradient Boosting classifier when combined with attitude, social norm and perceived behavioral control factors, as evidenced by its highest accuracy and macro F1 score among the best-performing models. On the other hand, Naive Bayes consistently exhibited a satisfactory performance across all possible combinations of the theory of planned behavior.

## Confusion matrix

Fig 12 shows the confusion matrices for best-performing models in terms of the theory of planned behavior factors' possible combinations. Fig 12(a) from the confusion matrices performed better than others. Therefore, Gradient Boosting with combined Attitude, social Norm and Perceived behavioral Control factors performed the most promising. It was able to classify 21 HIGH classes and 19 LOW classes correctly. However, 3 HIGH classes were classified as Low and 1 LOW class as HIGH. Hence, 4 classes were miss classified, still the lowest among other models.

## Receiver operating characteristic curve

The Receiver Operating Characteristic (ROC) curve provides insights into the model's ability to discriminate between the positive and negative classes. Upon comparing the Receiver Operating Characteristic (ROC) curves depicted in Fig 13, it can be observed that the majority of the classifiers exhibit a slight variation in their performance. The area under ROC curve (AUC) values for the High and LOW classes exhibit a slight increase when utilizing the K-nearest neighbors classifier, as depicted in Fig 13(b), with a value of 0.96, surpassing the AUC

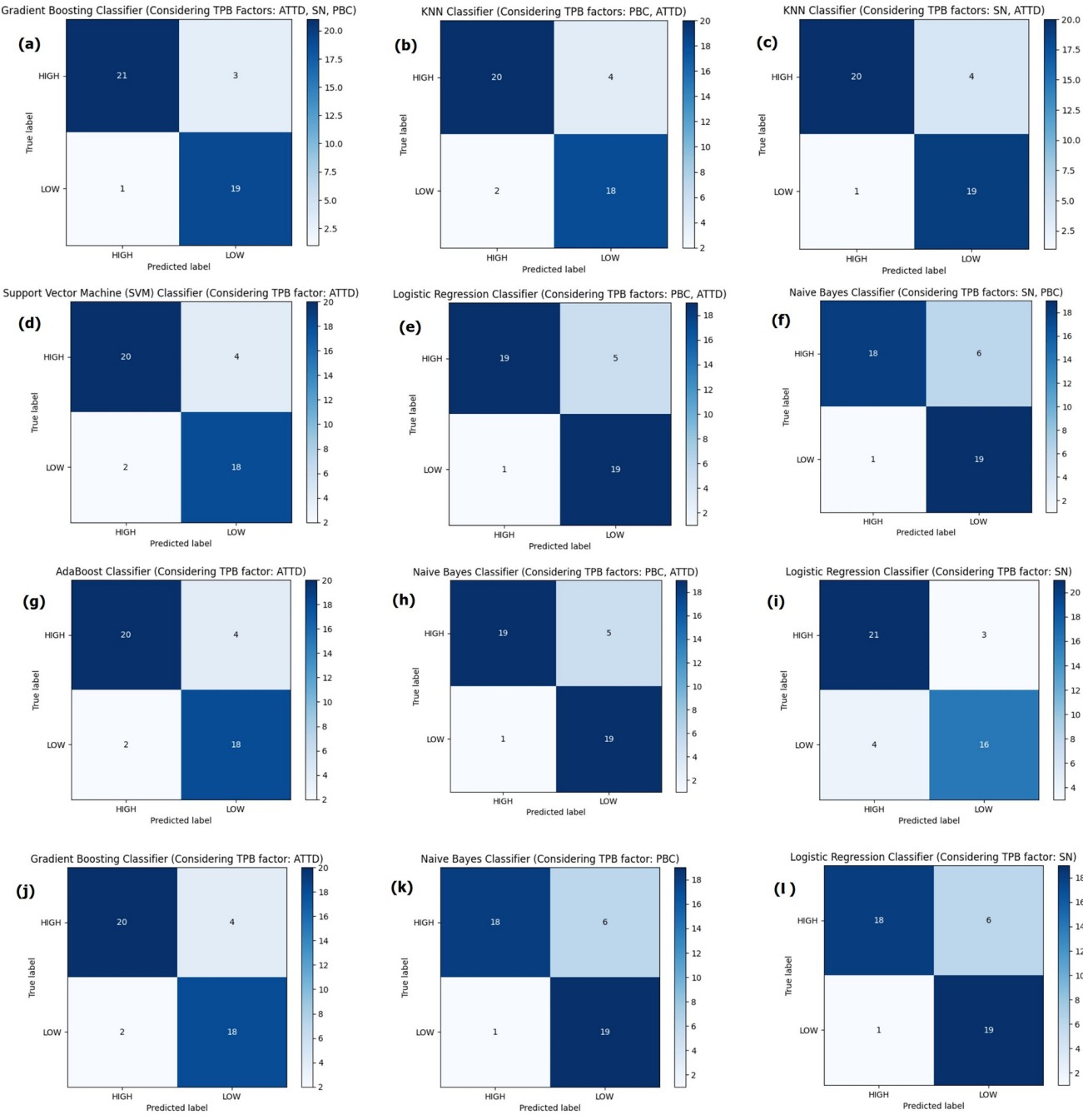

**Fig 12. Confusion matrix of the best performing model in each combination of TPB factors.**

values of other classifiers. The KNN Classifier demonstrates a marginally superior discriminative capacity.

## Precision recall curves

Fig 14 shows the precision-recall curve. It depicts the tradeoff between precision(Eq (6)) and recall(Eq (5)) for different thresholds. A high area under the curve indicates a combination of

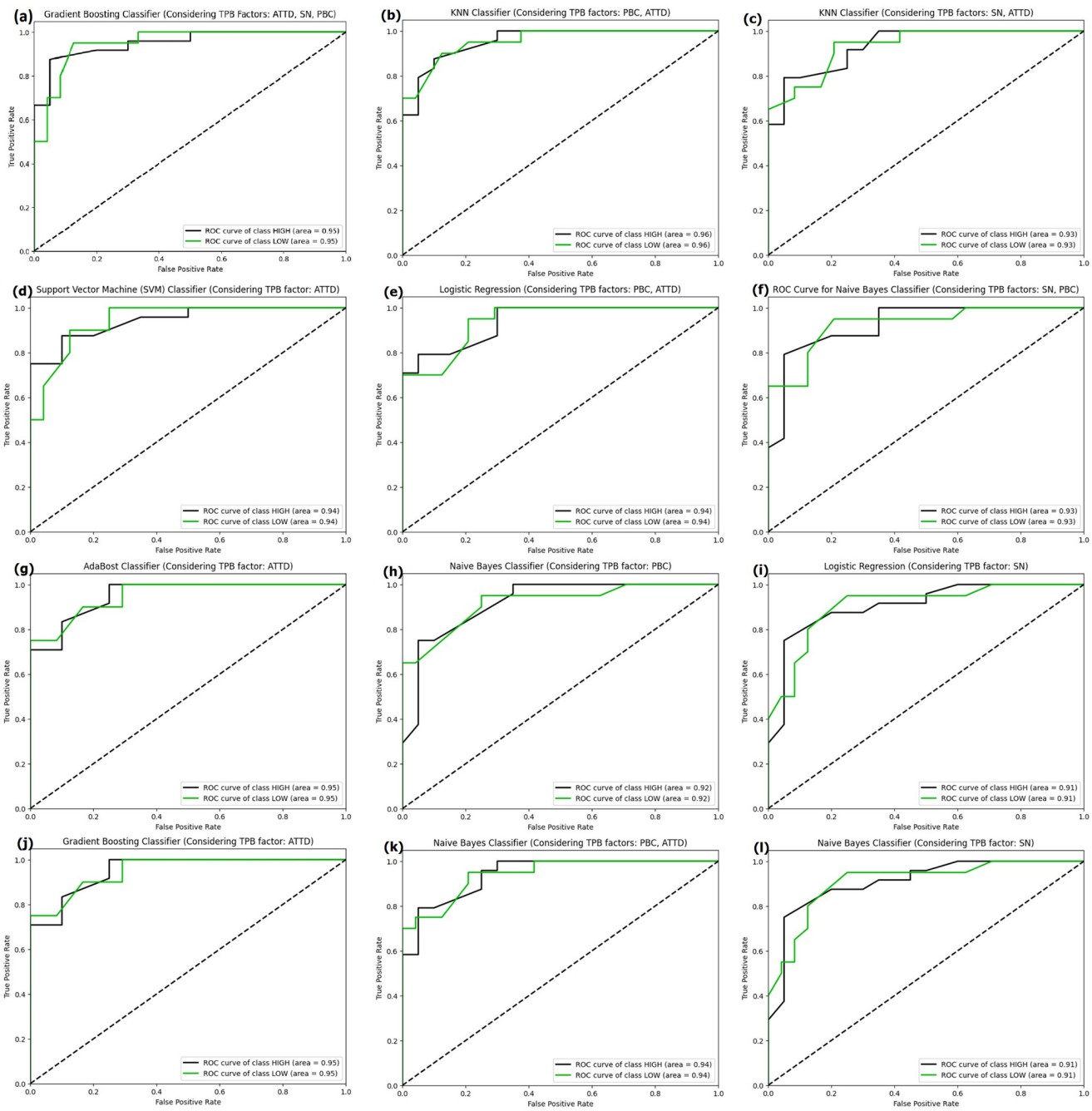

**Fig 13. ROC curves of the best performing models in each combination of TPB factors.**

high recall and high precision, with high precision indicating a low false positive rate(FPR) and high recall indicating a low false negative rate(FNR). For Fig 14(a) the area of precision-recall curve of HIGH class is 0.96 and LOW class is 0.94 indicating high true positive rate (TPR) and high true negative rate(TNR). Among the Fig 14(a)–14(l), Gradient Boosting combined with all the TPB factors (Fig 14(a)) performed well in terms of area under the curve (AUC) of precision recall curves.

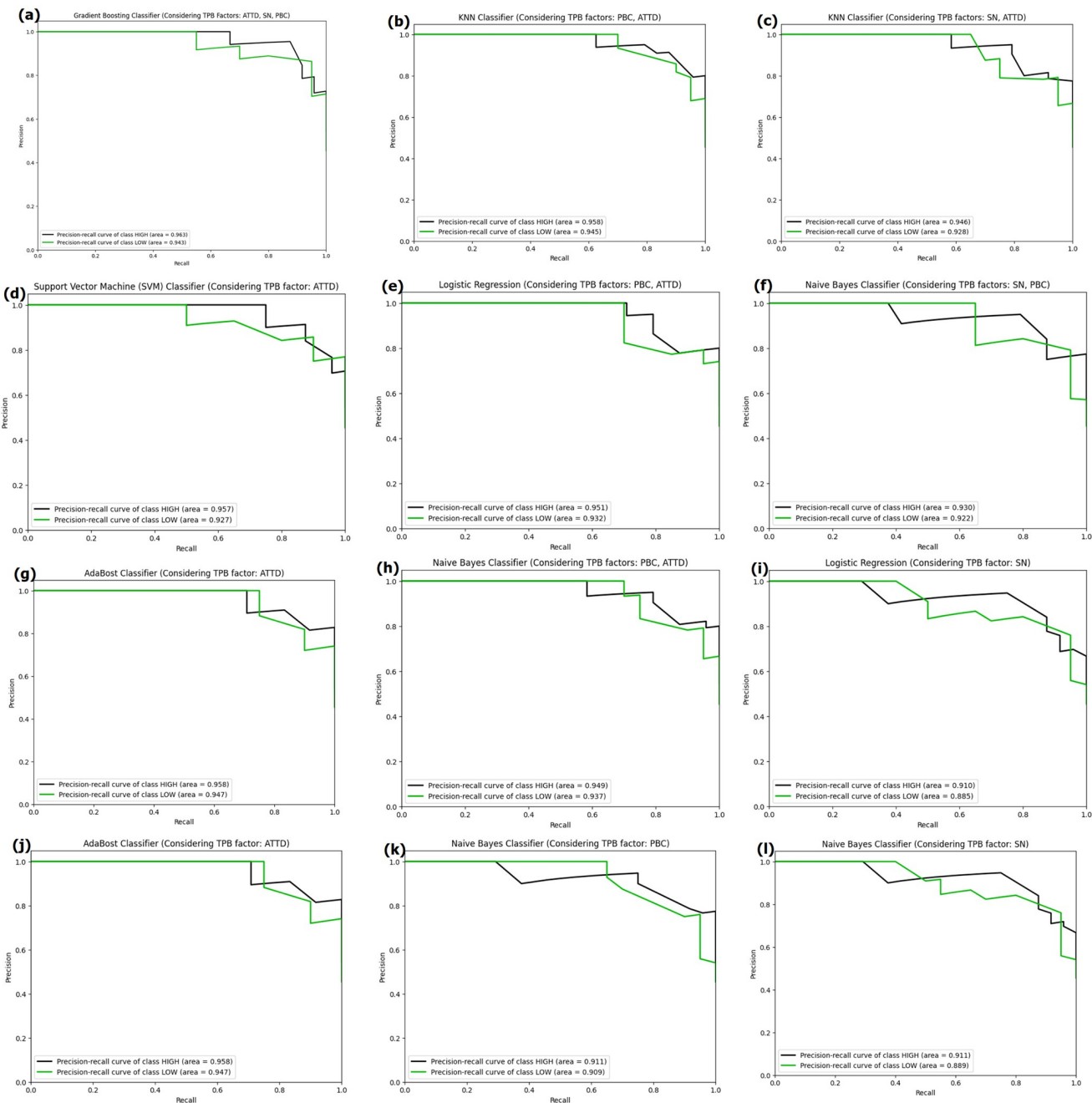

**Fig 14. Precision recall curves of the best performing models in each combination of TPB factors.**

## Explainable AI

LIME (Local Interpretable Model-Agnostic Explanations) provides a quick way to understand how black box machine learning models operate [43]. By constructing a simpler surrogate model that approximates the complex ML model at a local level, LIME enables the analysis of individual predictions within confined areas. It provides logical explanations within those specific regions. In the context of purchase behavior prediction, the LIME explainable AI

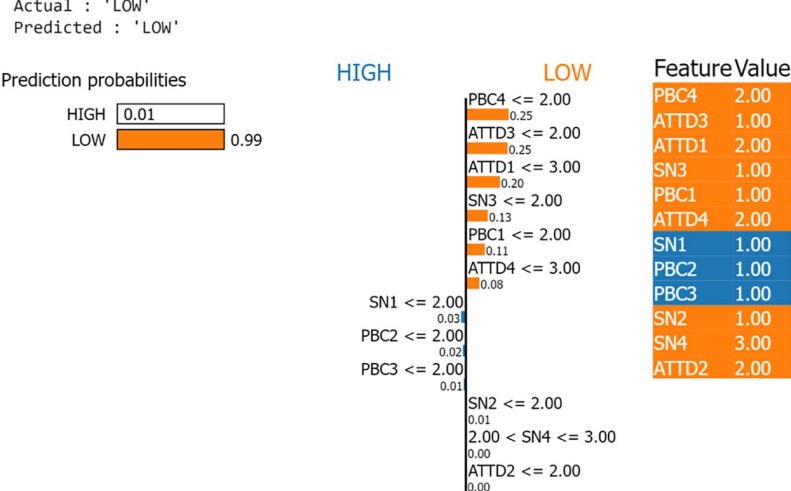

**Fig 15. The interpretation of purchase behavior prediction of low influence case using LIME explainable AI.**

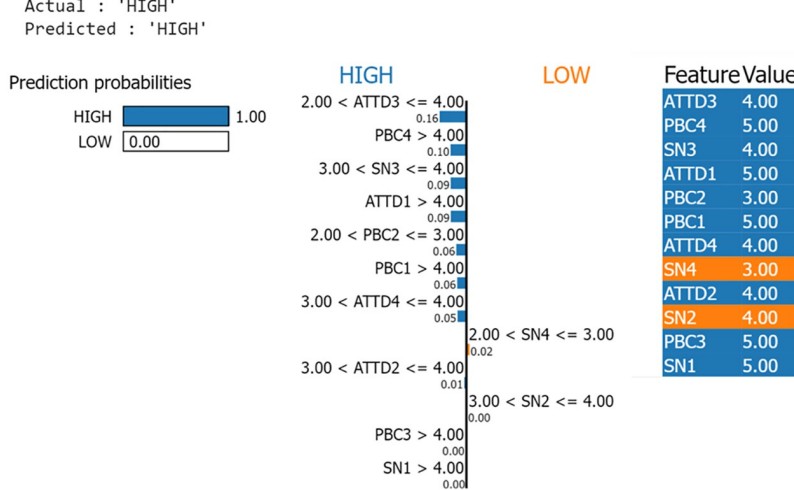

**Fig 16. The interpretation of purchase behavior prediction of high influence case using LIME explainable AI.**

framework reveals insightful interpretations for high and low instances, as depicted in Figs 15 and 16, respectively. In this particular example, the Gradient Boosting model, combined with the Theory of planned behaviors, confidently predicts low behavior influence with a score of 0.99 (as shown in Fig 15) and high behavior influence with a score of 1.0 (as shown in Fig 16). LIME identifies PBC4, ATTD3 and ATTD1 as the three prominent factors that decides the LOW class while ATTD3, PBC4 and SN3 as three prominent factors that decides the HIGH class.

## Research implications

Based on result analysis and Explainable AI insights, several managerial and social implications can be explored in order to assist businesses to excel.

## Managerial implications

The findings and understandings obtained via the utilization of Explainable AI in this study have significant management ramifications for businesses that operate on social media platforms. The Explainable AI analysis highlights the significant role of variables like ATTD1, ATTD3, and PBC4 in determining the LOW class, indicating that individuals with lower Likert scores on these factors have reduced motivation to make online purchases. This highlights the importance for managers to consider these factors when making business decisions, offering organizations the opportunity to tailor product advertisements to specifically target individuals with lower purchasing inclinations. At the same time, these insights enable more efficient allocation of marketing budgets. Managers can strategically invest resources in customers exhibiting stronger purchasing inclinations, potentially yielding higher returns compared to those with weaker tendencies. The XAI analysis further identifies customers with elevated Likert ratings on factors like ATTD3, PBC4, SN3, and PBC1, allowing managers to pinpoint profitable clients and craft strategic marketing plans to effectively target them. This approach optimizes resource allocation, enhances overall business profitability, and bolsters competitiveness. Furthermore, with insights gained from predictive analytics, businesses can potentially identify new market segments or product niches with growth potential. This can open up opportunities for businesses to expand their product offerings and enter new markets, thereby increasing their revenue and market share.

## Social implications

The study has a number of possible social implications including:

1. **Improved customer engagement**: Customer data can potentially be used to gain insights about customer preferences, behaviors, and purchasing patterns. Organizations can use this data to individualize their engagements with customers, thus improving customer experience. Personalization, in turn, can result in higher customer satisfaction, increased loyalty, and repeat business, all of which are crucial for the growth of businesses as well as to the society.

2. **Economic impact in developing economies**: The expansion of businesses in developing economies often serves as a pivotal catalyst for economic development. The prosperity of businesses leads to the generation of employment opportunities, the stimulation of local economies, and also contribute towards poverty reduction. Predictive tools have the potential to expedite the expansion of businesses by augmenting their competitive edge and financial gains.

3. **Digital inclusion**: The integration of predictive analytics tools helps facilitate the transition of businesses into the digital era. As these tools become more accessible and affordable, even smaller businesses in developing economies can harness their power. This contributes to digital inclusion and can help bridge the digital divide in regions where technology adoption is still limited.

## Discussion

Despite the fact that the research was limited by a small sample size of 219 respondents, we were able to achieve a remarkable performance (Tables 7 and 8) through the use of the appropriate classifier, feature selection and hyperparameter optimization.

Our investigation yielded insightful insights into various facets of human nature, attitudes, and perspectives. A substantial majority of respondents agreed that social media had a

profound effect on them. Individuals discovered new products, encountered promotions, discounts, and advertisements, and ultimately made purchase decisions as a result of social media. Moreover, respondents were interested in recommending social media platforms to other users as effective marketplaces. These findings demonstrate the substantial impact and influence that social media has on human behavior.

By leveraging the power of predictive models, we can accelerate marketing strategies toward a more efficient and targeted paradigm, maximizing their impact and developing a stronger relationship between businesses and consumers.

## Conclusion

This paper describes various machine learning techniques and evaluates their predictions of consumer purchasing behavior. Combining Theory of Planned behavior with machine learning, this paper takes a psychological along with machine learning approach to predict the consumer purchasing behavior. Our research data set was obtained from the [15], which consists of responses to a questionnaire. We created seven distinct combinations of the three factors (attitudes, subjective norms, and perceived behavioral control) from the Theory of Planned behavior. Following that, we used eight different classifiers to predict on these seven combinations of factors, and found that the Gradient Boosting Classifier performed the best with a 91% accuracy and similar macro F1 score when all three factors were combined. Based on the ROC curve graph, KNN and Gradient Boosting Classifier had better class discriminative ability than others in terms of Area under the curve for respective classes. Furthermore, precision-recall curves (Fig 14) demonstrated that Gradient Boosting performed well in terms of both precision and recall. The article included an explainable AI explanation of LIME that demonstrates the effect of features on the final outcomes.

## Future scope

In future, we plan to deploy our model as an application software. This will enable enterprises to employ our model on their own data—thus providing a customized solution. The present study was conducted on a limited number of samples, which resulted in certain constraints on the extent and comprehensiveness of analysis. However, these limitations offer significant prospects for enhancing our study methodology. In future, we plan to collect large amount of data, which will improve the depth of analysis. In addition, future research endeavors may employ deep learning methodologies to analyze large datasets.

## Author Contributions

**Conceptualization:** Md. Shawmoon Azad, Sifat Momen.

**Formal analysis:** Md. Shawmoon Azad, Rezwan Hossain, Raiyan Rahman, Sifat Momen.

**Investigation:** Md. Shawmoon Azad, Shadman Sakib Khan, Rezwan Hossain, Raiyan Rahman, Sifat Momen.

**Methodology:** Md. Shawmoon Azad, Shadman Sakib Khan, Rezwan Hossain, Sifat Momen.

**Project administration:** Sifat Momen.

**Supervision:** Sifat Momen.

**Visualization:** Md. Shawmoon Azad, Raiyan Rahman, Sifat Momen.

**Writing – original draft:** Md. Shawmoon Azad, Shadman Sakib Khan, Rezwan Hossain, Raiyan Rahman, Sifat Momen.

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
