## [Decision Letter · Decision Letter 0]

14 Aug 2023

PONE-D-23-21252Predictive modeling of consumer purchase behavior on social media: integrating theory of planned behavior and machine learning for actionable insightsPLOS ONE

Dear Dr. Momen,

Thank you for submitting your manuscript to PLOS ONE. After careful consideration, we feel that it has merit but does not fully meet PLOS ONE’s publication criteria as it currently stands. Therefore, we invite you to submit a revised version of the manuscript that addresses the points raised during the review process.

We look forward to receiving your revised manuscript.

Kind regards,

Shanjida Chowdhury

Academic Editor

PLOS ONE

Journal Requirements:

Reviewers' comments:

Reviewer's Responses to Questions

**Comments to the Author**

1. Is the manuscript technically sound, and do the data support the conclusions?

Reviewer #1: Partly

2. Has the statistical analysis been performed appropriately and rigorously? 

Reviewer #1: Yes

3. Have the authors made all data underlying the findings in their manuscript fully available?

Reviewer #1: Yes

4. Is the manuscript presented in an intelligible fashion and written in standard English?

Reviewer #1: No

5. Review Comments to the Author

Reviewer #1: Abstract is structured well but it only talks about the findings in the way it is represented in the paper. It would be better if the abstract explains the outcome with implications.

Introduction should also address: Why the current research is required? How the research contributes to the existing body of knowledge?

Literature needs to be strengthened from TPB perspective. Why and How TPB is suitable for this study and how it has been used in past?

A strong recommendation is to highlight the literature gap.

Research methodology should address the following issues:

Clarity on study design

Sampling design

Area of study, which state/ region?

Method of data collection and why?

Common method bias

Non-response bias

Demographic profile of respondent (Table)

Data visualization can be done in a better way in terms of placement of charts/ figures.

A complete sub-section required for following:

Managerial Implication

Social Implication

Future research directions

6. PLOS authors have the option to publish the peer review history of their article (what does this mean?). If published, this will include your full peer review and any attached files.

Reviewer #1: No

---

## [Author Response · Author response to Decision Letter 0]

26 Sep 2023

Please see the pdf titled "Response to Reviewers" attached in the file section for responses to reviewers and editor comments.

---

## [Decision Letter · Decision Letter 1]

12 Dec 2023

Predictive modeling of consumer purchase behavior on social media: integrating theory of planned behavior and machine learning for actionable insights

PONE-D-23-21252R1

Dear Dr. Momen

We’re pleased to inform you that your manuscript has been judged scientifically suitable for publication and will be formally accepted for publication once it meets all outstanding technical requirements.

Kind regards,

Shanjida Chowdhury

Academic Editor

PLOS ONE

Additional Editor Comments (optional):

Reviewers' comments:

Reviewer's Responses to Questions

**Comments to the Author**

1. If the authors have adequately addressed your comments raised in a previous round of review and you feel that this manuscript is now acceptable for publication, you may indicate that here to bypass the “Comments to the Author” section, enter your conflict of interest statement in the “Confidential to Editor” section, and submit your "Accept" recommendation.

Reviewer #2: All comments have been addressed

2. Is the manuscript technically sound, and do the data support the conclusions?

Reviewer #2: Yes

3. Has the statistical analysis been performed appropriately and rigorously? 

Reviewer #2: Yes

4. Have the authors made all data underlying the findings in their manuscript fully available?

Reviewer #2: Yes

5. Is the manuscript presented in an intelligible fashion and written in standard English?

Reviewer #2: Yes

6. Review Comments to the Author

Reviewer #2: please put more explanation:

contribution of this research,

matching the aim with the conclusion.

and put more explanation why the reader need to read this paper, what the important things

7. PLOS authors have the option to publish the peer review history of their article (what does this mean?). If published, this will include your full peer review and any attached files.

Reviewer #2: **Yes: **Jacky Chin

---

## [Editor Report · Acceptance letter]

15 Dec 2023

PONE-D-23-21252R1 

PLOS ONE

Dear Dr. Momen, 

I'm pleased to inform you that your manuscript has been deemed suitable for publication in PLOS ONE. Congratulations! Your manuscript is now being handed over to our production team.

Kind regards, 

on behalf of

Dr. Shanjida Chowdhury 

Academic Editor

PLOS ONE